# Role of the Pfannenstiel Incision in Robotic Hepato-Pancreato-Biliary Surgery

**DOI:** 10.3390/jcm12051971

**Published:** 2023-03-02

**Authors:** Kosei Takagi, Yuzo Umeda, Ryuichi Yoshida, Tomokazu Fuji, Kazuya Yasui, Jiro Kimura, Nanako Hata, Takahito Yagi, Toshiyoshi Fujiwara

**Affiliations:** Department of Gastroenterological Surgery, Okayama University Graduate School of Medicine, Dentistry, and Pharmaceutical Sciences, Okayama 700-8558, Japan

**Keywords:** robotic surgery, minimally invasive surgery, hepato-pancreato-biliary surgery, Pfannenstiel incision

## Abstract

Studies remain limited on the role of the Pfannenstiel incision in minimally invasive hepato-pancreato-biliary (HPB) surgery, especially robotic surgery. The role of various extraction sites in robotic HPB surgery should be understood. Herein, we describe the surgical techniques, outcomes, advantages, and disadvantages of the Pfannenstiel incision in robotic pancreatic surgery. Seventy patients underwent robotic pancreatectomy at our institution between September 2020 and October 2022. The Pfannenstiel incision was used for specimen retrieval in 55 patients. Advantages of the Pfannenstiel incision include less pain, cosmetic benefits, and a lower incidence of complications. Moreover, the specimen could be removed using the robotic system docked. However, all complex reconstructions should be performed intra-abdominally during robotic pancreatoduodenectomies. The incidence of mortality and postoperative pancreatic fistula (grade B) was 0% and 9.1%, respectively. During the median follow-up (11.2 months) after surgery, complications at the Pfannenstiel incision site included surgical site infection (*n* = 1, 1.8%) and incisional hernia (*n* = 1, 1.8%). The Pfannenstiel incision can be a useful option for specimen retrieval in minimally invasive HPB surgery, according to the surgeon’s preferences and the patient’s condition.

## 1. Introduction

The Pfannenstiel incision, commonly used in gynecological and urological surgery, is reportedly associated with a lower incidence of wound complications, including surgical site infection and incisional hernia, than midline incisions [1,2]. Selecting the incision site for a large specimen retrieval is a major issue, particularly in minimally invasive hepato-pancreato-biliary (HPB) surgery; a transumbilical incision is widely used for specimen removal during laparoscopic HPB surgery. To date, few studies have reported the impact of the Pfannenstiel incision on minimally invasive HPB surgery [3]. However, our protocol for robotic HPB surgery includes the Pfannenstiel incision for specimen retrieval, owing to several advantages [4,5]. Although some HPB surgeons are unfamiliar with this incision, surgeons should understand the differences in specimen retrieval sites between laparoscopic and robotic surgery. Moreover, the role of various extraction sites in robotic HPB surgery should be understood.

Herein, we aimed to describe the surgical techniques and outcomes, as well as the advantages and disadvantages of the Pfannenstiel incision, particularly focusing on robotic pancreatic surgery.

## 2. Materials and Methods

### 2.1. Patients and Data Collection

Seventy patients underwent robotic pancreatectomy, including 40 pancreatoduodenectomies and 30 distal pancreatectomies, at our institution between September 2020 and October 2022. Of the 70 patients, 55 who underwent specimen retrieval via the Pfannenstiel incision were included in this study.

Using a prospectively collected database, we collected the following clinical data: age, sex, body mass index, primary disease, type of procedure, operative time, blood loss, conversion to open surgery, mortality, postoperative pancreatic fistula grade according to the International Study Group of Pancreatic Surgery (ISGPS) [6], postoperative hospital stay, and complications at the Pfannenstiel incision site (surgical site infection and incisional hernia).

### 2.2. Overviews of Robotic Pancreatectomy

The surgical protocols and strategies used for robotic pancreatectomy in this study have been previously reported [4,5,7,8]. The da Vinci Si or Xi system (Intuitive Surgical, Sunnyvale, CA, USA) was used. Trocar placement during robotic pancreatectomy is shown in Figure 1. Total intravenous general anesthesia was administered without epidural anesthesia. The specimen was removed between resection and reconstruction in pancreatoduodenectomy and after resection in distal pancreatectomy via the Pfannenstiel incision with the robotic system (robotic arms) docked. Note that during Pfannenstiel incision placement, the robotic instruments should be removed to avoid incidental organ injury.

### 2.3. Surgical Technique of the Pfannenstiel Incision

A 6–7 cm transverse incision was made approximately two finger widths cephalad to the pubic bone (Figure 2a). Then, the subcutaneous tissues were dissected to expose the anterior rectus sheath (Figure 2b). The fascia was opened and separated from the rectus muscle superiorly and inferiorly, as wide as possible (Figure 2c). A second incision, which was vertical, was made to separate the rectus muscle and open the peritoneum (Figure 2d). Next, an endobag was inserted, and the specimen was extracted (Figure 2e).

Following the removal of the specimen, the peritoneum was immediately closed with a continuous suture (Figure 2f). The rectus muscle was re-approximated in the middle using interrupted sutures to close the rectus diastasis (Figure 2g), and a continuous suture was placed close to the anterior rectus sheath (Figure 2h). Finally, the skin was closed with subcuticular sutures. The Pfannenstiel incision technique during robotic pancreatoduodenectomy is presented in Appendix A.

### 2.4. Advantages and Disadvantages of the Pfannenstiel Incision

The advantages and disadvantages of the Pfannenstiel incision during robotic HPB surgery are summarized in Table 1. Advantages include less pain, cosmetic benefits, and a lower incidence of complications. Moreover, the specimen could be extracted via the Pfannenstiel incision with the robotic system docked (Figure 3). However, the primary disadvantage is that an additional incision is required. Moreover, all complex reconstructions, including pancreaticojejunostomy, hepaticojejunostomy, and gastrojejunostomy, should be performed intra-abdominally during robotic pancreatoduodenectomies.

## 3. Results

The characteristics and outcomes of the 55 patients are summarized in Table 2. There were 30 men and 25 women, with a median (interquartile range, IQR) age of 72 (range, 60–75) years. The most common indication for surgery was pancreatic ductal adenocarcinoma (*n* = 16, 29.1%), followed by duodenal (*n* = 12, 21.8%) and ampullary (*n* = 9, 16.4%) tumors.

The overall incidence rates of mortality and postoperative pancreatic fistula (grade B) were 0% and 9.1%, respectively. The median (IQR) of postoperative hospital stay was 9 (range, 8–12) days. During the median follow-up (11.2 months) after surgery, Pfannenstiel-related complications included surgical site infection (*n* = 1, 1.8%) and incisional hernia (*n* = 1, 1.8%).

## 4. Discussion

The present study demonstrated the role of the Pfannenstiel incision in robotic HPB surgery. It is crucial to understand the surgical techniques, advantages, and disadvantages of the Pfannenstiel incision as an option for specimen retrieval in robotic HPB surgery.

The Pfannenstiel incision has several advantages, including better healing and greater strength than vertical incisions, resulting in a lower tendency for fascial dehiscence and hernia formation [9]. Moreover, retrieving specimens via the Pfannenstiel incision is beneficial in terms of cosmesis and reduced pain [10]. Even though the Pfannenstiel incision is widely used in gynecological and urological surgery, it should also be considered in minimally invasive HPB surgery, especially when a wide incision for the retrieval of large specimens is required.

In laparoscopic hepatectomy, a transumbilical incision is often made by extending the navel port site craniocaudally for specimen retrieval [11]. Although the Pfannenstiel incision is not commonly used in minimally invasive HPB surgery, comparable outcomes between Pfannenstiel and midline incisions have been reported in laparoscopic hepatectomy [3,12]. However, the differences between laparoscopic and robotic surgeries should be emphasized. In robotic surgery, the robotic system must be undocked when retrieving the specimen through a transumbilical incision. Conversely, the Pfannenstiel incision allows the robotic system to be docked during specimen retrieval (Table 1).

A recent meta-analysis reported that the incidence of incisional hernia and surgical site infection at Pfannenstiel incision sites was 2.1% and 11.0%, respectively [13]. Moreover, the incidence of incisional hernia was lower at the Pfannenstiel incision site than at other extraction sites, such as midline and umbilical incisions [13]. In the present study, the incidence of surgical site infection and incisional hernia at the Pfannenstiel incision site was 1.8% and 1.8%, respectively. These results were acceptable; however, further studies are required.

Several other techniques have also been developed. For example, the use of a small-sized wound protector may be helpful as a modified technique for Pfannenstiel incisions. Instead of closing the Pfannenstiel incision immediately, it allows for quick pneumoperitoneum restoration and can be closed subsequently. A small-sized wound protector can be maintained until the end of the surgery and might help check every trocar site for bleeding at removal. In contrast, the Kustner incision could be chosen instead of Pfannenstiel. Although there could be a few advantages of the Kustner incision in terms of febrile morbidity and wound infection [14], it might be associated with a higher incidence of incisional hernias and requires a larger incision compared to the Pfannenstiel incision [15]. Therefore, further evidence is required to confirm the role of the Kustner incision.

This study had several limitations. Although the surgical techniques of robotic pancreatectomy have been standardized at our institution, the findings were based on our limited experience using a retrospective analysis with a small sample size and a short follow-up period. As evidence of the efficacy of the Pfannenstiel incision in minimally invasive HPB surgery remains limited, further studies with larger sample sizes and long-term follow-ups are required. Moreover, this is a note and not an original article. Therefore, this study lacked a control group, and the outcomes were compared with a group treated with other retrieval incisions.

## 5. Conclusions

The Pfannenstiel incision can be a useful option for specimen retrieval in minimally invasive HPB surgery, depending on the surgeon’s preferences and the patient’s condition. Surgeons should understand the advantages and disadvantages of various extraction sites and select the most suitable one for robotic HPB surgery.

## Figures and Tables

**Figure 1 jcm-12-01971-f001:**
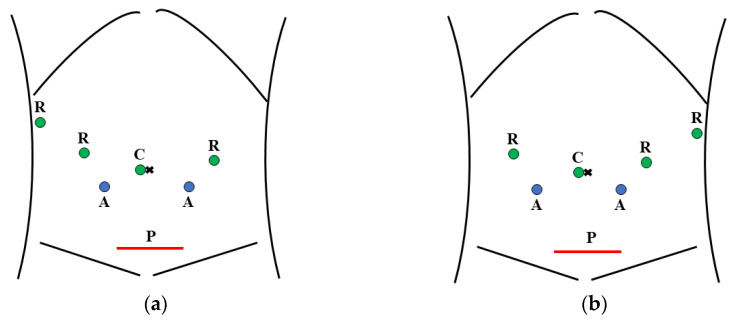
Trocar and Pfannenstiel incision placements during robotic pancreatectomy. (**a**) Robotic pancreatoduodenectomy; (**b**) Robotic distal pancreatectomy. R, robotic trocar; C, camera; A, trocar for an assistant; P, Pfannenstiel incision.

**Figure 2 jcm-12-01971-f002:**
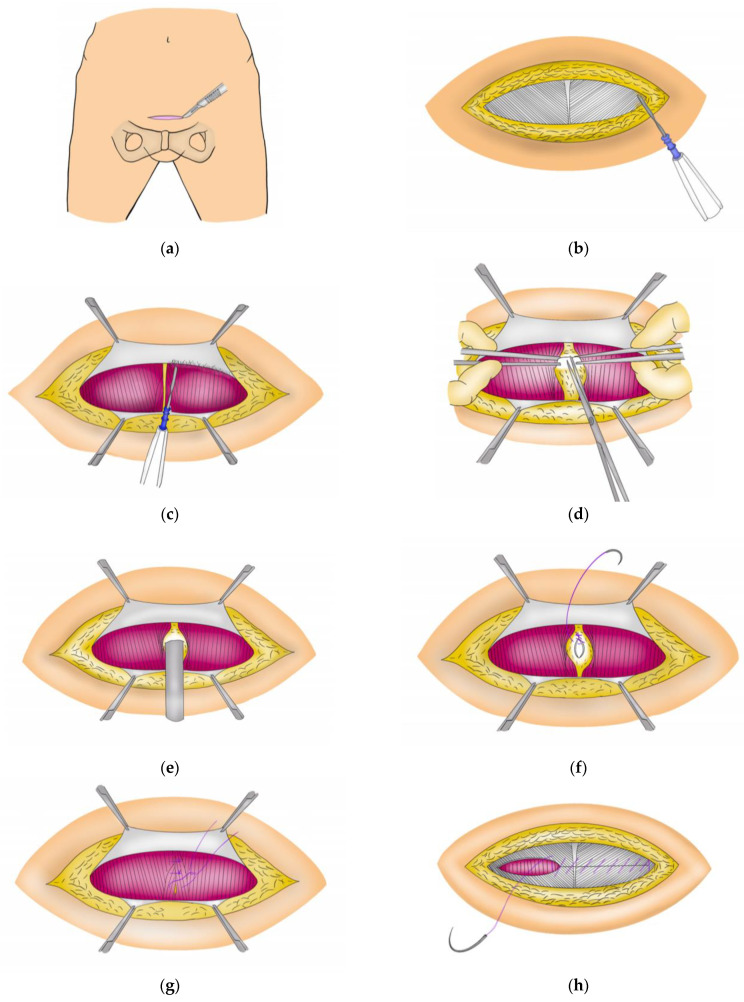
The Pfannenstiel incision technique. (**a**) a 6–7 cm transverse incision; (**b**) exposure of the anterior rectus sheath; (**c**) separation of the fascia from the rectus muscle; (**d**) opening the peritoneum; (**e**) insertion of an endobag and removal of the specimen; (**f**) closing the peritoneum; (**g**) re-approximation of the rectus muscle; and (**h**) closing the anterior rectus sheath.

**Figure 3 jcm-12-01971-f003:**
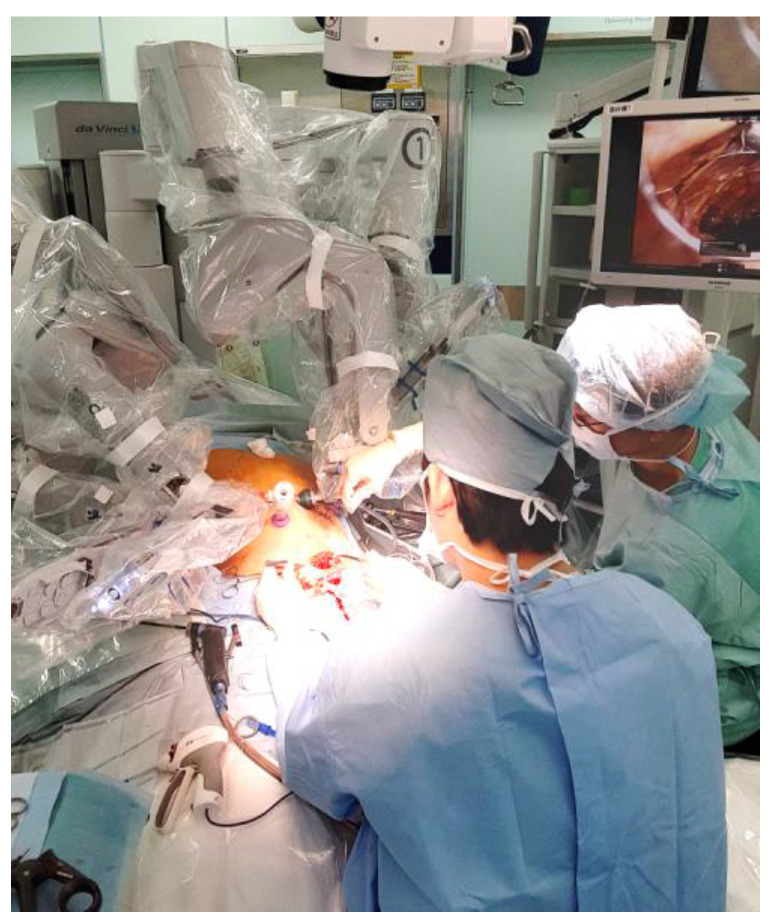
Placement of the Pfannenstiel incision during robotic pancreatoduodenectomy. The specimen was removed via the Pfannenstiel incision with the robotic system docked.

**Table 1 jcm-12-01971-t001:** Advantages and disadvantages of the Pfannenstiel and transumbilical incision in robotic pancreatectomy.

	Advantages	Disadvantages
Pfannenstiel incision	Less painful Cosmetic benefit Lower risk of wound complications Lower risk of incisional hernia The specimen can be removed with the robotic system docked	Additional incision is required Can be difficult in patients with adhesions All reconstructions should be performed intraabdominally during pancreatoduodenectomy
Transumbilical incision	Reconstructions can be performed through the incision during pancreatoduodenectomy	More painful The robotic system needs to be undocked

**Table 2 jcm-12-01971-t002:** Clinical data of patients who underwent specimen retrieval via the Pfannenstiel incision during robotic pancreatectomy at our institution between September 2020 and October 2022.

Variables	Patients (*n* = 55)
Preoperative factors	
Age, year	72 (60–75)
Gender	
Male	30 (54.5)
Female	25 (45.5)
Body mass index, kg/m^2^	23.0 (21.4–25.4)
Primary diseases	
Pancreatic ductal adenocarcinoma	16 (29.1)
Duodenal tumor	12 (21.8)
Ampullary tumor	9 (16.4)
Intraductal papillary mucinous neoplasm	5 (9.1)
Bile duct tumor	5 (9.1)
Pancreatic neuroendocrine tumor	4 (7.3)
Others	4 (7.3)
Operative factors	
Type of procedure	
Robotic pancreatoduodenectomy	*n* = 39
Operative time, minutes	406 (384–478)
Blood loss, mL	70 (10–100)
Robotic distal pancreatectomy	*n* = 16
Operative time, minutes	245 (227–271)
Blood loss, mL	50 (0–115)
Conversion to open surgery	0 (0)
Postoperative factors	
Mortality	0 (0)
Postoperative pancreatic fistula (grade B)	5 (9.1)
Postoperative hospital stays, days	9 (8–12)
Complications at the Pfannenstiel incision	
Surgical site infection	1 (1.8)
Incisional hernia	1 (1.8)
Follow-up, months	11.2 (7.9–17.9)

Data are presented as median (interquartile range) or number (percentage).

## Data Availability

Not applicable.

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
