# Peer review of "Role of the Pfannenstiel Incision in Robotic Hepato-Pancreato-Biliary Surgery"

_jcm, 2023, doi:10.3390/jcm12051971_

Round 1
Reviewer 1 Report
This is an interesting technical note about the role of Pfannenstiel's incision in retrieving the specimen in HPB robotic surgery. The Pfanennestiel's incision is a consolidated retrieval method already well used in colorectal minimally invasive laparoscopic and robotic surgery. I understand the opportunity to explain its role in pancreatic surgery. Anyway, this is a note, not an original article. The authors should review their discussion section to smooth their position and discuss the lack of a control group and the missing comparison of the outcomes with a group treated with other retrieval incisions.
Author Response
Dear Editor:
RE: jcm-2192790
The Role of the Pfannenstiel Incision in Robotic Hepato-Pancreato-Biliary Surgery
Thank you for reviewing our manuscript. We are pleased that our manuscript was favorably reviewed and found to be potentially acceptable for publication pending revisions.
We thank you and the reviewers for your thoughtful suggestions and insights. The manuscript has benefited from these insightful suggestions. I look forward to working with you and the reviewers to move this manuscript closer to publication in Journal of Clinical Medicine.
The manuscript has been rechecked and the necessary changes have been made in accordance with the reviewers’ suggestions. The responses to all comments have been prepared and attached herewith.
Thank you for your consideration. I look forward to hearing from you.
Sincerely,
On behalf of all coauthors,
Kosei Takagi, MD, PhD
Department of Gastroenterological Surgery, Okayama University Graduate School of Medicine, Dentistry, and Pharmaceutical Sciences, 2-5-1 Shikata-cho, Kita-ku, Okayama 700-8558, Japan
Tel: +81-86-223-7151; Fax: +81-86-221-8775; E-mail: kotakagi15@gmail.com
Reviewer 1
Comment 1:
This is an interesting technical note about the role of Pfannenstiel's incision in retrieving the specimen in HPB robotic surgery. The Pfanennestiel's incision is a consolidated retrieval method already well used in colorectal minimally invasive laparoscopic and robotic surgery. I understand the opportunity to explain its role in pancreatic surgery. Anyway, this is a note, not an original article. The authors should review their discussion section to smooth their position and discuss the lack of a control group and the missing comparison of the outcomes with a group treated with other retrieval incisions.
Response:
Thank you for your positive feedback. As reviewer 1 pointed out, this is a note, not an original article, with the lack of a control group. In the revised manuscript, we have described this issue in the limitation session (page 6, line 154-156).
Reviewer 2 Report
Thanks to the authors for sharing their experience on such topic in robotic surgery.
I have few questions/comments:
1) Do you have any experience with the Kustner incision instead of classical Pfannenstiel? I agree on implementing suprapubic incisions in hepato-biliary robotic surgery when the whole procedure is carried out robotically but maybe Kustner could provide even less discomfort, wound infections and hematomas comparing with Pfannenstiel because you avoid aponeurosis detachment from the underlying rectus muscle and reduce aponeurotic lateral extension of the incision (corners far from midline are usually reported by patients as more painful). Would you comment on that in the text, citing literature about the Kustner approach?
2) Have ever tried to insert a small-sized wound protector (for example Alexis-S) once you open the peritoneum? Instead of closing the Pfannenstiel incision right away , it allows a quick PNP restoration and can be close later. It is handy, can be kept until the end of the surgery, mantaining PNP, and it is useful to check every single trocar site at extraction , preventing bleeding.
3) With Pfannenstiel it is possible, as you stated, to avoid daVinci undocking, can you explain more clearly in the text if you remove the robotic instruments (not arms) from the abdominal cavity when you fashion the suprapubic incision?
Thank you in advance for submitting such manuscript and for addressing the above questions.
Author Response
Dear Editor:
RE: jcm-2192790
The Role of the Pfannenstiel Incision in Robotic Hepato-Pancreato-Biliary Surgery
Thank you for reviewing our manuscript. We are pleased that our manuscript was favorably reviewed and found to be potentially acceptable for publication pending revisions.
We thank you and the reviewers for your thoughtful suggestions and insights. The manuscript has benefited from these insightful suggestions. I look forward to working with you and the reviewers to move this manuscript closer to publication in Journal of Clinical Medicine.
The manuscript has been rechecked and the necessary changes have been made in accordance with the reviewers’ suggestions. The responses to all comments have been prepared and attached herewith.
Thank you for your consideration. I look forward to hearing from you.
Sincerely,
On behalf of all coauthors,
Kosei Takagi, MD, PhD
Department of Gastroenterological Surgery, Okayama University Graduate School of Medicine, Dentistry, and Pharmaceutical Sciences, 2-5-1 Shikata-cho, Kita-ku, Okayama 700-8558, Japan
Tel: +81-86-223-7151; Fax: +81-86-221-8775; E-mail: kotakagi15@gmail.com
Reviewer 2
Thanks to the authors for sharing their experience on such topic in robotic surgery. I have few questions/comments:
Comment 1:
Do you have any experience with the Kustner incision instead of classical Pfannenstiel? I agree on implementing suprapubic incisions in hepato-biliary robotic surgery when the whole procedure is carried out robotically but maybe Kustner could provide even less discomfort, wound infections and hematomas comparing with Pfannenstiel because you avoid aponeurosis detachment from the underlying rectus muscle and reduce aponeurotic lateral extension of the incision (corners far from midline are usually reported by patients as more painful). Would you comment on that in the text, citing literature about the Kustner approach?
Response:
Thank you for your feedback. We have no experiences of the Kustner incision instead of classical Pfannenstiel. However, the Kustner incision could be optional instead of the Pfannenstiel incision. Although there could be a few advantages of the Kustner incision in terms of febrile morbidity and wound infection, the Kustner incision might be associated with higher incidence of incisional hernias, and require larger incision compared to the the Pfannenstiel incision. Therefore, further evidence should be required to confirm the role of the Kustner incision. In the revised manuscript, we have discussed on this issue in the discussion session (page 6, line 143-148).
Comment 2:
Have ever tried to insert a small-sized wound protector (for example Alexis-S) once you open the peritoneum? Instead of closing the Pfannenstiel incision right away, it allows a quick PNP (pneumoperitoneum) restoration and can be close later. It is handy, can be kept until the end of the surgery, mantaining PNP, and it is useful to check every single trocar site at extraction, preventing bleeding.
Response:
Thank you for your feedback. Although we have no experiences of using a small-sized wound protector, we agree with the reviewer 2’s opinions. In the revised manuscript, we have discussed on this issue as a modified technique for Pfannenstiel incision (page 6, line139-143).
Comment 3:
With Pfannenstiel it is possible, as you stated, to avoid daVinci undocking, can you explain more clearly in the text if you remove the robotic instruments (not arms) from the abdominal cavity when you fashion the suprapubic incision?
Response:
Thank you for your feedback. During the Pfannenstiel incision placement, we keep the robotic system (robotic arms) docked, but remove robotic instruments to avoid incidental organ injury. In the revised manuscript, we have added comments on this issue more clearly (page 2, line 61-62).
Round 2
Reviewer 1 Report
Thank you for your corrections. The article improved. Still I think that the self citations should be limited. Moreover, english language should ne reviewed.
Author Response
Thank you for reviewing our manuscript. We are pleased that our manuscript was favorably reviewed and found to be potentially acceptable for publication pending revisions. We thank you and the reviewers for your thoughtful suggestions and insights. The manuscript has benefited from these insightful suggestions. I look forward to working with you and the reviewers to move this manuscript closer to publication in Journal of Clinical Medicine.
The manuscript has been rechecked and the necessary changes have been made in accordance with the reviewers’ suggestions. The responses to all comments have been prepared and attached herewith.
Thank you for your consideration. I look forward to hearing from you.
Sincerely,
On behalf of all coauthors,
Kosei Takagi, MD, PhD
Reviewer 1 Comment 1: Thank you for your corrections. The article improved. Still I think that the self citations should be limited. Moreover, english language should ne reviewed.
Response: Thank you for your feedback. As reviewer 1 pointed out, we have limited the self citations. In the revised manuscript, we have cited only important manuscripts. Moreover, the revised manuscript has been checked by Editage (www.editage.jp) for English language editing.